# Synthesis of cBN-hBN-SiC_w_ Nanocomposite with Superior Hardness, Strength, and Toughness

**DOI:** 10.3390/nano13010037

**Published:** 2022-12-22

**Authors:** Lei Sun, Yitong Zou, Mengdong Ma, Guangqian Li, Xiaoyu Wang, Xiang Zhang, Zewen Zhuge, Bing Liu, Yingju Wu, Baozhong Li, Zhisheng Zhao

**Affiliations:** 1Center for High Pressure Science (CHiPS), State Key Laboratory of Metastable Materials Science and Technology, Yanshan University, Qinhuangdao 066004, China; 2Macao Institute of Materials Science and Engineering, Macau University of Science and Technology, Taipa, Macao 999078, China

**Keywords:** cBN, SiC_w_, high-pressure sintering, toughening mechanism

## Abstract

Nanocomposites with one-dimensional (1D) and two-dimensional (2D) phases can demonstrate superior hardness, fracture toughness, and flexural strength. Cubic boron nitride-hexagonal boron nitride-silicon carbide whiskers (cBN-hBN-SiC_w_) nanocomposites with the simultaneous containing 1D SiC_w_ and 2D hBN phases were successfully fabricated via the high-pressure sintering of a mixture of SiC_w_ and cBN nanopowders. The hBN was generated in situ via the limited phase transition from cBN to hBN. Nanocomposites with 25 wt.% SiC_w_ exhibited optimal comprehensive mechanical properties with Vickers hardness of 36.5 GPa, fracture toughness of 6.2 MPa·m^1/2^, and flexural strength of 687.4 MPa. Higher SiC_w_ contents did not significantly affect the flexural strength but clearly decreased the hardness and toughness. The main toughening mechanism is believed to be a combination of hBN inter-layer sliding, SiC_w_ pull-out, crack deflection, and crack bridging.

## 1. Introduction

Cubic boron nitride (cBN) has a wide range of industrial applications owing to its outstanding hardness, thermal conductivity, and chemical stability [1,2]. What is far more important is the ability of cBN to cut ferrous and carbide-forming hard substances where diamond completely fails [3]. Although single cBN crystals can be prepared under industrial pressure (≤6 GPa), the samples grown by high-pressure methods are size-limited to the millimeter scale [4]. In addition, the anisotropy of single cBN crystals brings inconvenience to subsequent processing and use. For the pure-phase polycrystalline cBN, whose mechanical properties even exceed those of single-crystal cBN, the successful synthesis of dense samples cannot be achieved without harsh pressure conditions (≥7.7 GPa), whether through the high-pressure phase transformation of hexagonal boron nitride (hBN), amorphous boron nitride, onion boron nitride (oBN) or the high-pressure sintering of cBN [5,6,7,8,9].

cBN-based composites with metal or ceramic additives that are prepared under mild temperature and pressure conditions (≤6 GPa) with good performance are currently attracting extensive attention and undergoing rapid development [10,11]. Accordingly, two kinds of additives were extensively studied to fabricate cBN-based composites. One is metal additives of the groups IV, V, and VI of the periodic table and their compounds or other metallic elements such as aluminum, cobalt, and nickel [12,13,14]. Although metal additives can effectively reduce the sintering pressure and temperature, they adversely affect the mechanical properties of the composites. The other is ceramic additives with high hardness, high modulus, and high wear resistance, such as Al_2_O_3_ [15], SiAlON [16], and Y_2_O_3_-stabilized ZrO_2_ (YSZ) [17]. Among a multitude of ceramics additives, SiC as an efficient reinforcement material for cBN-based composites has received considerable attention because of its outstanding properties. The hardness of 80 wt.% cBN-15 wt.% TiC-5 wt.% SiC composites fabricated at 1350 °C under 5.5 GPa is 516.3 MPa, which is higher than the 224.9 MPa of the 95 wt.% cBN-5 wt.% Ti_3_SiC_2_ composite [18]. The cBN-AlN-Si_3_N_4_-SiC composites with PSN and Al as additives were fabricated by high-pressure sintering (5 GPa, 1450 °C). The sintered sample with the addition of 30 wt.% PSN and 10 wt.% Al showed a high hardness of 25.2 GPa, which is corresponding to a 26% increase compared with the value for the sample with the addition of 30 wt.% Al [19]. Thus, the type of additive has a direct impact on the mechanical properties of cBN-based composites, such as hardness, fracture toughness, flexural strength, and wear resistance.

Recently, low-dimensional ceramic materials have been the most widely chosen additives to fabricate high-performance composites [20,21,22,23,24,25]. One-dimensional (1D) materials involve nanofibers [20,26,27], nanotubes [21,22,28], and whiskers [23,24,25,29], which are singular structures with the lateral dimension in submicron or nanometer scale. One-dimensional materials are ideal systems for exploring various new phenomena at the nanoscale and studying the size and dimensionality dependence of functional properties. The addition of 1D ceramic materials can not only ensure the hardness and strength of composites but also overcome the intrinsic brittleness of ceramic composites. Taking this into account, SiC whiskers (SiC_w_) have been widely used as an additive for diverse ceramic-based composites. Whisker-reinforced composites of ZrB_2_-SiC_w_ [20], Al_2_O_3_-SiC_w_ [23], TiC-SiC_w_ [30], SiC-SiC_w_ [31], and Si_3_N_4_-SiC_w_ [32] were reported. It was found that the fracture toughness of TiB_2_-SiC_w_ composites was 7.8 MPa·m^1/2^, which is approximately 130% higher than that of the single TiB_2_ phase ceramic [33]. Two-dimensional (2D) materials are defined as crystalline materials consisting of single- or few-layer atoms, in which the in-plane interatomic interactions are much stronger than those along the stacking direction. Two-dimensional materials represented by graphene have been proven to be an ideal reinforcement to fabricate ceramic composites due to their large specific surface, high aspect ratio, and superior mechanical, thermal and electrical properties [34]. It was reported that the fracture toughness of Si_3_N_4_-graphene composites and B_4_C-graphene composites were substantially improved up 6.6 MPa·m^1/2^ and 8.76 MPa·m^1/2^, which is 135% and 131% higher than that of the corresponding pure-phase ceramics, respectively [35,36]. Similar to graphene, hBN has also shown great promise as a reinforcement and has been used in several ceramic composites [37,38,39,40,41,42]. It was found that fracture toughness and flexural strength of B_4_C-10 vol.% hBN composite were 4.2 MPa·m^1/2^ and 494 GPa, which were increased by ~75% and ~40% compared to the B_4_C ceramic in the absence of hBN, respectively [41]. Therefore, the introduction of appropriate 1D and 2D materials into cBN is conducive to improving the comprehensive performance of composites. Nevertheless, to the best of our knowledge, there are few reports on cBN composites with the simultaneous addition of 1D and 2D materials.

In this work, novel cBN-hBN-SiC_w_ nanocomposites were successfully fabricated via the high-pressure sintering of SiC_w_ and cBN powder. The hBN was generated in situ by a limited transformation of cBN to hBN. The effects of the SiC_w_ on the phase composition and mechanical properties of nanocomposites were systematically investigated, and the toughening mechanism of the cBN-hBN-SiC_w_ nanocomposites was also analyzed.

## 2. Materials and Methods

The commercial powders including cBN particles (particle size of ~600 nm, >99.9% purity, InnoChem Science & Technology Co., Ltd., Beijing, China), and SiC whiskers (diameter in the range of 0.2–1.2 μm, length in the range of 50–100 μm, Qinhuangdao Eno High-Tech Material Development Co., Ltd., Qinhuangdao, China) were used as starting materials in our work. SiC_w_ was mixed with cBN via alcohol using an agate mortar and pestle for about 1 h, and the scope of SiC_w_ contents in mixtures is 0–30 wt.%. To get rid of impurities, the mixtures were treated in a vacuum (3.0 × 10^−5^ Pa) at 800 °C for 0.5 h and then quickly pre-pressed into a cylindrical pellet with Φ (diameter) = 5 mm × 5 mm and placed into an hBN crucible in order to get rid of the isolated the air. DS 6 × 8 MN cubic press machine (Guilin Guiye Machinery Co., Ltd., Guilin, China) was used to accomplish the high-pressure sintering experiments. The mixtures were slowly cold-compressed to 6 GPa and then heated to 1450 °C at a rate of 200 °C/min for 20 min. For each SiC_w_ content, we prepared three samples. The cBN-hBN-SiC_w_ composites with 0, 5, 10, 15, 20, 25, and 30 wt.% SiC_w_ were denoted as SC0, SC5, SC10, SC15, SC20, SC25, and SC30, respectively. The surface of the sample was wet-ground with 320-, 600-, and 800-grit silicon carbide paper. The ground surfaces were further polished on a polishing machine using 3-, 1-, and 0.5-micron-sized diamond paste. The nanomaterials are cylindrical blocks (Φ 4.5 mm × 4.5 mm) after grinding and polishing.

The phase compositions of the powders and the nanocomposites were detected using X-ray diffraction (XRD, Smartlab Rigaku, Tokyo, Japan) at 40 kV and 40 mA with Cu Kα radiation (λ = 1.5406 Å), the scanning step width of 0.02°, and the rate of 1°/min, in the 2θ range from 20° to 70°. The morphology and fracture surface of nanocomposites were characterized by scanning electron microscope (SEM, Verios G4 UC, Thermo Fisher Scientific, Waltham, MA, USA) with through-lens detector (TLD), and the chemical composition was analyzed by energy dispersive X-ray spectroscopy (EDS, Ultim Extreme, Oxford Instruments, Abingdon, UK). The microstructure characterizations of the nanocomposites were examined by transmission electron microscopy (TEM, JEM-ARM200F2, JEOL Ltd., Tokyo, Japan) at an accelerating voltage of 200 kV. The specimens of nanocomposites for TEM analysis were prepared by ion milling using a Ga-focused ion beam (FIB, Scios Dual beam, Thermo Fisher Scientific, Waltham, MA, USA) at an accelerating voltage of 30 kV. The thick plates were initially pre-cut to 5 μm by using a current of 30 nA. Final thinning to electron-transparent slices (thickness of less than 100 nm) was carried out by a FIB machine with ion beam current from 9.1, 2.5, and 0.79 to 0.19 nA. The Vickers hardness (*H_V_*) was measured by a hardness testing machine (KB-5-BVZ, Rems-Murr, Germany) under an indentation load of 9.8 N (adopted loading time = 30 s; dwell time = 15 s). The flexural strength of specimens with dimensions of 0.5 mm (width) × 1.0 mm (height) × 3.2 mm (length) was measured by the three-point bending method using a mechanical test machine (MTII/Fullman-SEMtester 2000, MTI Instruments, Inc., Albany, NY, USA) with a 2.45 mm testing span and a 0.5 mm/min crosshead speed. The fracture toughness (*K_IC_*) was calculated by assessing the Vickers indentation cracks with a load of 9.8 N, according to the Evans equation [43]: *K_IC_* = 0.16*H*·*a*^2^*c*^−1.5^(1)
where 0.16 is the calibration constant, *H* is the Vickers hardness (GPa), *a* is the half diagonal of the indent (μm), and *c* is the radial crack length (μm).

## 3. Results and Discussion

Figure 1a indicates the XRD patterns of cBN, SiC_w,_ and cBN-25 wt.% SiC_w_ mixed powders after vacuum treatment at 800 °C. The diffraction peaks detected in the XRD pattern of mixed powders at 43.314°, and 50.430°, correspond to the (111) and (200) reflections of cBN (PDF No. 35-1365), respectively. The diffraction peaks corresponding to (111) and (220) reflections of SiC (PDF No. 29-1129), are located at 35.597° and 59.977°, respectively. The weak peak at 33.6° in XRD patterns of mixed powders can be attributed to the stacking faults of (111) planes in SiC, similar to previous observations [44]. It is worth noting that no peaks of hBN (PDF No. 34-0421) were detected. Figure 1b presents the SEM images of the raw material of SiC_w_, and they have regular shapes and a mean diameter of ~420 nm based on the SEM images (Figure 1b). A typical SEM image shows that the cBN powders also have a uniform particle size (Figure 1c), based on which the grain size of cBN was evaluated. In order to ensure accuracy, the number of randomly selected cBN grains is more than 200. The average particle size is calculated as 592 nm. As shown in Figure 1d, SiC_w_ is evenly dispersed in the cBN powders after grinding and mixing.

Figure 2a shows the X-ray diffraction patterns of the as-sintered cBN-hBN-SiC_w_ nanocomposites with different SiC_w_ content. The phases detected by XRD are SiC, hBN, and cBN in the nanocomposites, indicating that there is a limited transformation from cBN to hBN and no reaction between cBN and SiC_w_. As shown in the STEM image of the SiC25 (Appendix A), a very reduced amount of hBN was observed, which was in good agreement with the results analyzed by XRD. Figure 2b shows a typical SEM image on the polished surface of the SiC25. The SiC_w_ were broken into short rods (white area) after high-pressure sintering with unchanged morphologies. The corresponding EDS compositional maps (Figure 2c) further confirmed that the distribution of SiC_w_ is highly uniform and the arrangement is random in the BN matrix.

The Vickers hardness, fracture toughness, and flexural strength of the cBN-hBN-SiC_w_ nanocomposites with various SiC_w_ content are presented in Figure 3. Due to the formation of low-hardness hBN, the Vickers hardness of SiC0 (27.1 GPa) is lower than the value of the single cBN crystal and pure-phase polycrystalline cBN [5,6,7,8,9]. However, the fracture toughness of the nanocomposites is higher than that of the single cBN crystal (~2.8 MPa·m^1/2^) [7]. Previous work also confirmed the toughening effect of hBN [41,45]. As displayed in Figure 3, as the content of SiC_w_ increases, the Vickers hardness of cBN-hBN-SiC_w_ nanocomposites initially increases and then reaches an inflection point. The same trend is also observed for fracture toughness. SiC is not a superhard material as with cBN, and it still has the inherent brittleness of ceramics. Thus, the excessive introduction of SiC_w_ leads to a decrease in the Vickers hardness and fracture toughness for the SiC30. The flexural strength of the cBN-hBN nanocomposites is 486.4 MPa. With the addition of SiC_w_, the flexural strength of cBN-hBN-SiC_w_ nanocomposites increased to more than 600 MPa. Differing from the change trend of hardness and toughness, the further increase in SiC_w_ content has no obvious effect on the flexural strength. It is worth mentioning that the as-sintered SiC25 exhibited the optimal mechanical properties with the highest hardness of 36.5 ± 0.5 GPa, highest fracture toughness of 6.2 ± 0.6 MPa·m^1/2^, and flexural strength of 687.4 ± 44.9 MPa.

To comprehensively evaluate the mechanical properties of cBN-hBN-SiC_w_ nanocomposites, the hardness, fracture toughness, and flexural strength of cBN composites with various additives reported previously are listed in Table 1 [4,24,29,45,46,47,48,49]. As shown in Table 1, metal additives have the effect of toughening, but they cause a serious loss of hardness for cBN composites [14,46,47]. When introducing ceramic additives, the comprehensive mechanical properties of the cBN composites are obviously better than those prepared by using only metal additives [14,24,29,47,49]. The mechanical properties of cBN composites can be further improved by introducing 2D materials such as Ti_3_AlC_2_ and hBN [45,48], and 1D materials such as SiC whiskers and Al_2_O_3_ whiskers [4,24]. For cBN-hBN-SiC_w_ nanocomposites with the simultaneous containing 1D and 2D phases, the hardness and toughness of the SiC25 reach the best-reported level while the flexural strength is obviously better than the reported values [4,24,29,45,46,47,48,49].

SEM images of fracture surfaces for the nanocomposite with different SiC_w_ contents were shown in Appendix A. As shown in Appendix A, the fracture mode of cBN was an intergranular fracture. The fracture mode of the BN matrix was not affected by the addition of SiC_w_ (Appendix A). In addition, for samples with different contents of SiC_w_, SiC_w_ were uniformly distributed in the matrix. An SEM image of the fracture morphology of the as-sintered SiC25 is shown in Figure 4a. The fracture mode can be determined as typical intergranular fracture due to the presence of numerous individual submicron grains with regular facet shapes and intact fracture surfaces. Traces of whisker pull-out along the axial direction and whisker breakage can also be clearly observed, which results in an increase in fracture toughness and flexural strength. Such a phenomenon can be further confirmed by the SEM images of the radial crack of the Vickers indentation, as displayed in Figure 4b,c. The crack propagates a zigzag path at the nanoscale, along which many nanoscale crack-deflection and grain-bridging phenomena can be clearly observed. It is interesting to note that the radial microcrack from the indentation bypasses the SiC_w_ arranged axially on the polishing surface but passes through the SiC_w_ arranged radially. It is worth noting that these crack-deflection and grain-bridging phenomena can dissipate a large amount of energy, thus improving the fracture toughness.

The detailed microstructure of the as-sintered SiC25 was further investigated by TEM analysis. Figure 5a is a representative scanning transmission electron microscopy (STEM) image of the as-sintered nanocomposites. Obviously, hBN with a thickness of ~30 nm is between cBN grains, which could be identified by the corresponding interplanar spacings (Figure 5b–d). The relatively low stress in the small pores after cold pressing might be responsible for the hBN appearing. After cold pressing, the small pores are in a state of low pressure as compared with the macroscopic pressure or average stress throughout the whole sample. As the sample temperature increased, atoms in the grain surface around pores would be out of the stable region of the cBN phase and transferred to the hBN phase. Therefore, hBN did not exist the same structural aspect over the whole surface at the interface (Figure 5e). As shown in Figure 5e–h, the interplanar spacings of 0.209 nm, 0.252 nm, and 0.311 nm corresponding to (111) plane of cBN, (111) plane of SiC, and (0002) plane of the hBN, show good agreement with the corresponding standard PDF cards (Appendix A), respectively. The generated hBN by in situ phase transformation ensures that there is a uniform distribution of 2D materials between ceramic grains, which is a remarkable advantage over other mixing methods. The soft-phase hBN with a lamellar structure uniformly distributed among ceramic particles, on the one hand, bonds ceramic particles to promote densification; on the other hand, it absorbs energy during crack propagation, thus improving the fracture toughness and strength [35,37]. Figure 5i shows the interfacial structure between cBN and SiC_w_, and it can be observed that cBN and SiC form a good bonding interface, which implies that cBN and SiC_w_ were well sintered and there is no amorphous or impurity phase between cBN and SiC_w_. Figure 5j, k reveals strong hBN interfacial adhesion with SiC and cBN. Strong interfacial characteristics among the three phases serve to improve the strength and toughness of the composites. Therefore, cBN-based nanocomposites with a uniform distribution of SiC_w_ and hBN exhibit superior mechanical properties.

## 4. Conclusions

In summary, cBN-hBN-SiC_w_ nanocomposites with the simultaneous introduction of 1D and 2D phases were fabricated by the high-pressure sintering of mixed cBN and SiC_w_. The hBN was generated in situ via the phase transition from cBN to hBN. Composites with 1D SiC_w_ and 2D hBN phases can demonstrate superior hardness, fracture toughness, and flexural strength. The nanocomposite with 25 wt.% SiC_w_ exhibited optimal comprehensive mechanical properties with Vickers hardness of 36.5 GPa, fracture toughness of 6.2 MPa·m^1/2,^ and flexural strength of 687.4 MPa. The main toughening mechanism is believed to be a combination of hBN inter-layer sliding, SiC_w_ pull-out, crack deflection, and crack bridging. Our work demonstrates the preparation of nanocomposites by introducing multiple phases through various means to obtain multiple beneficial effects and may provide a reference and guidance for other subsequent research on high-performance composites.

## Figures and Tables

**Figure 1 nanomaterials-13-00037-f001:**
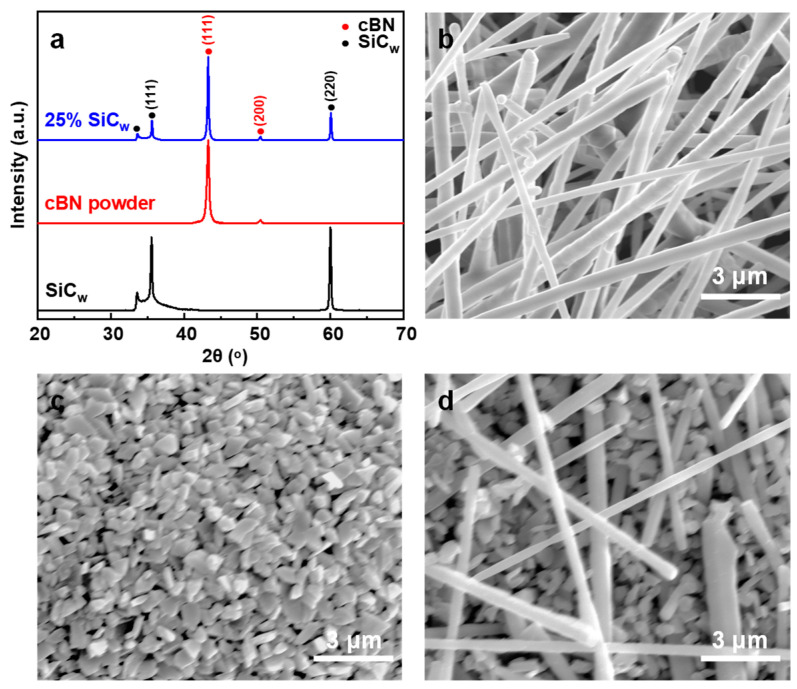
(**a**) XRD patterns of cBN, SiC_w,_ and cBN-25 wt.% SiC_w_ mixed powder after vacuum treatment. SEM images of the (**b**) SiC_w_, (**c**) cBN powders, and (**d**) cBN-25 wt.% SiC_w_ mixed powder.

**Figure 2 nanomaterials-13-00037-f002:**
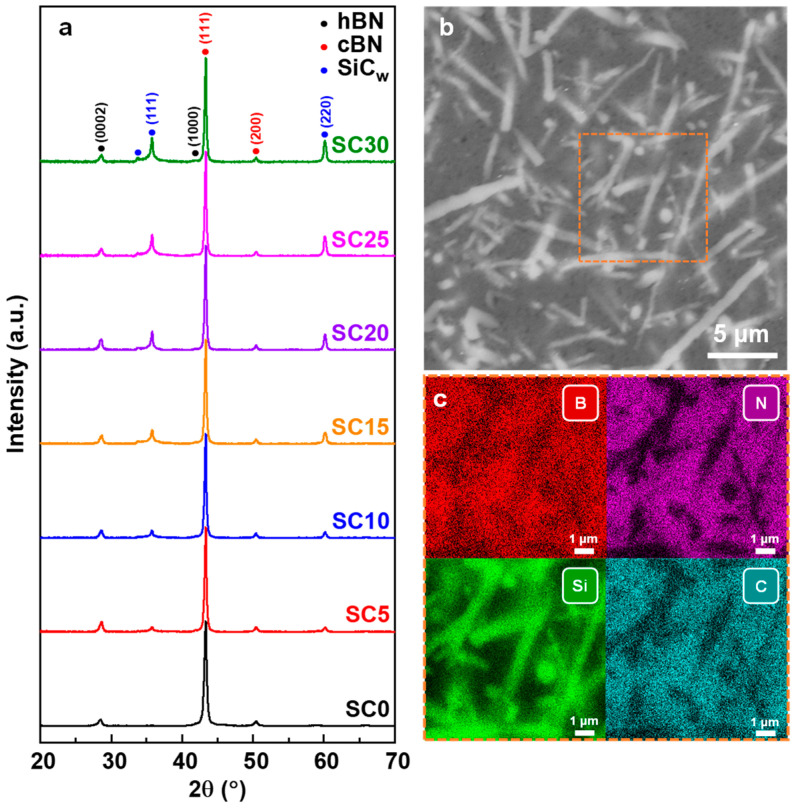
(**a**) XRD patterns of the cBN-hBN-SiC_w_ nanocomposites with different SiC_w_ content. (**b**) SEM image of the mirror-polished nanocomposite surface for cBN-hBN-SiC_w_ nanocomposite. (**c**) EDS mappings of the mirror-polished nanocomposite surface.

**Figure 3 nanomaterials-13-00037-f003:**
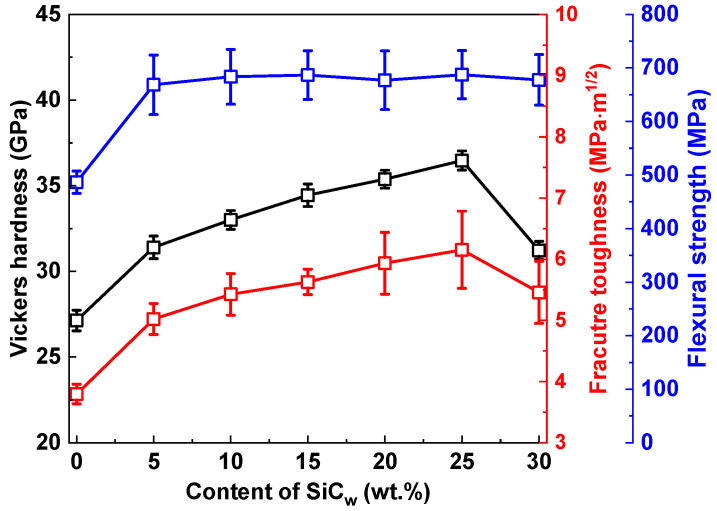
Vickers hardness, fracture toughness, and flexural strength of the cBN-hBN-SiC_w_ nanocomposites as a function of the content of SiC_w_.

**Figure 4 nanomaterials-13-00037-f004:**
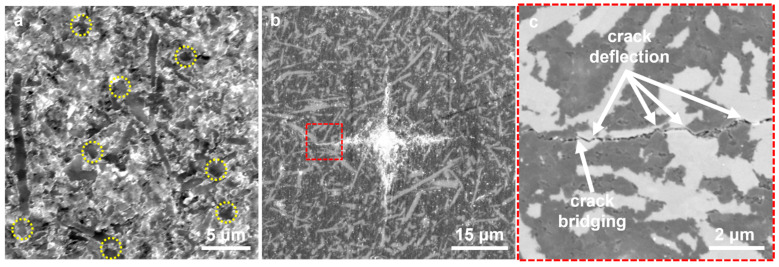
(**a**) SEM image of fracture surface for the SiC25. The whisker pull-out along axial direction is marked by the yellow circles. (**b**) SEM image of Vickers hardness indentation initiated by a load of 9.8 N for the SiC25. (**c**) High magnification of (**b**) indicates the presence of the crack deflection and crack bridging.

**Figure 5 nanomaterials-13-00037-f005:**
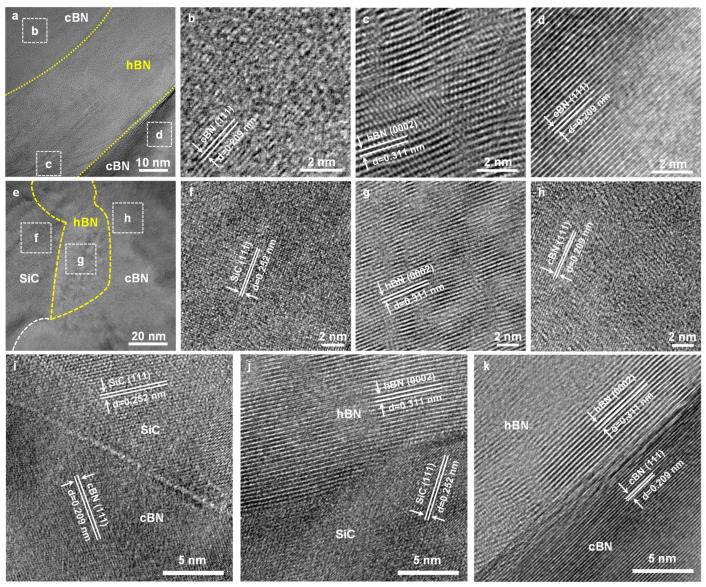
(**a**) Typical HRTEM image of cBN and hBN interface. (**b**–**d**) Magnified images of white solid box in (**a**). (**e**) Typical HRTEM image of the cBN-hBN-SiC_w_ interfaces. (**f**–**h**) Enlarged images of the white solid box in (**e**). HRTEM images of (**i**) cBN-SiC_w_ interface, (**j**) hBN-SiC_w_ interface, and (**k**) cBN-hBN interface.

**Table 1 nanomaterials-13-00037-t001:** Hardness, fracture toughness, and flexural strength of cBN-hBN-SiC_w_ nanocomposites in comparison with previously reported cBN composites with various additives.

Sample	Raw Materials	Hardness (GPa)	Fracture Toughness (MPa·m^1/2^)	Flexural Strength (MPa)	Reference
cBN composites with metal additives	95 wt.% cBN-5 wt.% Al	30.23	N/A	455.39	[47]
	45 vol.% cBN-35 vol.% Ti-20 vol.% Al	14.1	7.6 ^a^	390.7	[46]
	45 wt.% cBN-40 wt.% Ti-15 wt.% Al	14.14	3.95 ^b^	194.31	[14]
cBN composites with metals and ceramics as additives	60 vol.% cBN-25 vol.% NbN-5 vol.% Al-10 vol.% Al_2_O_4w_	27.5	5.5 ^c^	N/A	[24]
	50 vol.% cBN-5 vol.% Al-15 vol.% SiC_w_-30 vol.% TaN	33.01	6.57 ^c^		[29]
	50 vol.% cBN-5 vol.% Al-10 vol.% Al_2_O_3w_-35 vol.% TaN	33.09	2.59 ^c^		
cBN composites with ceramic additives	50 wt.% cBN-5 wt.% ZrO_2_ (3Y)-45 wt.% Al_2_O_3_	14.83	3.52 ^b^	145	[14]
	50 wt.% cBN-22 wt.% Si_3_N_4_-14 wt.% AlN-4 wt.% Y_2_O_3_-10 wt.% Al_2_O_3_	15.59	5.62 ^b^	465	
	90 wt.% cBN-10 wt.% TiC	29.02	N/A	497.06	[47]
	80 wt.% cBN-20 wt.% Ti_3_AlC_2_	33.14	N/A	422.4	[48]
	50 vol.% cBN@SiO_2_-15 vol.% TiN-35 vol.% TiB_2_	17.9	7.3 ^d^	N/A	[49]
	40 vol.% cBN-10 vol.% hBN-50 vol.% Al_2_O_3_	21.43	5.83 ^c^	N/A	[45]
cBN composites with ceramics and whiskers as additives	85 wt.% cBN-10 wt.% Al_3_BC_3_-5 wt.% SiC_w_	35	5.75 ^b^	316	[4]
	80 wt.% cBN-10 wt.% Al_3_BC_3_-10 wt.% SiC_w_	38	5.91 ^b^	345	
	75 wt.% cBN-10 wt.% Al_3_BC_3_-15 wt.% SiC_w_	38.2	6.31 ^b^	365	
	70 wt.% cBN-10 wt.% Al_3_BC_3_-20 wt.% SiC_w_	42.7	6.52 ^b^	406	
cBN-hBN-SiC_w_ nanocomposites	SiC25	36.5	6.2 ^e^	687.4	This work

^N/A^ indicates “not available”. ^a^ Calculated using Solozhenko equation (simplified Evans equation) based on Vickers hardness indentation. ^b^ Measured by single-edge-notched beam method. ^c^ Calculated using Anstis equation based on Vickers hardness indentation. ^d^ Calculated using Niihara equation based on Vickers hardness indentation. ^e^ Calculated using Evans equation based on Vickers hardness indentation.

## Data Availability

The data that support the findings of this study are available from the corresponding writer upon reasonable demand.

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
