# Peer review of "Synthesis of cBN-hBN-SiCw Nanocomposite with Superior Hardness, Strength, and Toughness"

_nanomaterials, 2022, doi:10.3390/nano13010037_

Round 1
Reviewer 1 Report
You need to state the size of your diamond polish, it does come in more than one size.
Reviewer 2 Report
Referee Report
“Synthesis of cBN-hBN-SiCw nanocomposite with superior hardness, strength, and toughness”
by Lei Sun, Yitong Zou, Mengdong Ma, Guangqian Li, Xiaoyu Wang, Xiang Zhang, Zewen Zhuge, Bing Liu, Yingju Wu, Baozhong Li and Zhisheng Zhao
submitted to Nanomaterials
This interesting paper devoted to preparing cBN-hBN-SiCw nanocomposites by introducing 1D and 2D additives to obtain multiple beneficial effects and may be a reference and guidance for other subsequent research of high-performance composites. Nanocomposite with 25 wt.% SiCw exhibited excellent comprehensive mechanical properties with Vickers hardness of 36.5 GPa, fracture toughness of 6.2 MPa·m1/2 and flexural strength of 687.4 MPa. The authors suggest that the main toughening mechanism is a combination of hBN inter-layer sliding, SiCw pull-out, as well as crack deflection and crack bridging. I liked this work, because the authors really achieved success and synthesized a composite with excellent mechanical properties. There is no doubt about the experience of the authors in their work on this topic. But in this article, I recommend making a few improvements to make it worthy of the high scientific level that is good for the Nanomaterials.
So, I think the article has good potential and can be published in Nanomaterials after minor revision.
Comments
1. I do not agree that additives can be called 1D and 2D, since they have dimensions of the micrometer order. Usually, only nanomaterials are called 1D and 2D, in which the dimensions are negligibly small.
2. In general, I ask you to explain why it was decided to call the obtained materials nanocomposites.
3. Please indicate the type of scanning electron microscope detector that was used. Surprisingly good contrast was obtained in the nanocomposite images.
4. the surprisingly small error in the determination of hardness must be explained. The material is multiphase, and indentation size is not much larger than the characteristic phase size. Please explain how such good accuracy was achieved.
5. My main commetn is the insufficient substantiation of the conclusion about the hardening mechanism. The authors suggest that the main toughening mechanism is a combination of hBN inter-layer sliding, SiCw pull-out, as well as crack deflection and crack bridging. But this was almost not confirmed by the microstructure. The microstructure is presented for only one sample. the explanation lies precisely in the change in the microstructure and in the behavior of the phases with a change in their concentration [10.3390/nano12121998, 10.1016/j.ijmecsci.2021.106952, 10.1016/j.surfcoat.2017.04.005, 10.1016/j.vacuum.2018.07.017, 10.3390/nano10061077]. I recommend adding microstructure images for all samples, not just the most successful one, and doing an analysis of the size distribution and behavior of the additives distributed in the matrix. Please be guided and take into account the articles recommended above.
6. The authors provided a Ttable 1 to compare the mechanical properties of similar materials. However, some references (22, 27) contain results obtained by a different method, which cannot be compared. Please check.
Reviewer 3 Report
The present paper is entitled «Synthesis of cBN-hBN-SiCw nanocomposite with superior hardness, strength, and toughness». The paper deals with the fabrication and the characterization of different nanocomposites by XRD, SEM, TEM and mechanical tests. The paper si suitable for publication in MDPI Nanomaterials, but major modifications are required.
Comments:
1. USE OF ENGLISH. English should be improved because it is not appropriate for a journal publication. Typos, poor syntax and other kind of mistakes are present. Here you are alist which is not exhaustive, but in any case representative:
lines 25-26: «What is far more important is the ability» a comme before "is" is required
line 29: «In addition, anisotropy of single cBN crystals bring inconvenience» it is "brings"
lines 32-33: «whether through the high-pressure phase...» please, improve the English of this sentence
lines 35-37: «cBN-based composites... is currently attracting...» the verb should be "are", because the the subject is singular
line 51: «composites of ZrB2-SiCw, etc.» "of" is not required here.
line 53: «TiB2 ceramic in the absence of» "the" is not needed here
line 69: «by high-pressure sintering SiCw and cBN powder. And the hBN was in-situ introduced by» between "sintering" and "SiCw", an "of" should be inserted; Also, the sentence starting with "And" should be improved.
line 77: «qinhuangdao, China» some capital letter should be used here.
lines 77-78: «SiCw with different contents (5 wt.%–30 wt.%) were mixed with cBN by alcohol with an agate mortar for 1 h.» this sentence should be improved
lines 156-157: «The mechanical properties of cBN composites» please, replace all of this with "They"
line 182: «which» is referred to h/BN? c/BN? please, improve the English
lines 193-194: «Strong interfacial characteristics among three phases benefit to improve the strength and toughness of composites.» This sentence is not correct, please, improve the English
line 207: « by high-pressure sintering mixed cBN and SiCw.» at least, a preposition "of" should be inserted between "sintering" and "mixed".
2. GENERAL COMMENTS. The statement of novelty of the information provided with the present paper needs to be revised, and particularly evidenced, because it is not clear what this work is bringing in terms of new results and improvements of the existing literature.
Please, in the Introduction section, provide more accurate details about 1D and 2D describing their features and properties. This is especially needed, because in the table 1 a lot of other phases are introduced (metals, oxides, etc.)
In general, the figures are quite smaller compared to the need of the reader: characters are barely visible, and reading is not easy.
It is not clear why the different conditions (= sample chemical compositions) were not named in a simple way, respective of their chemical composition; for example, SC25 for samples containing a 25 wt. % of SiC/w, etc.
3. SPECIFIC COMMENTS.
Title: the whole names of cBN-hBN-SiCw (cubic boron nitride-hexagonal boron nitride and silicon carbide whiskers) should be presented in the title, and the acronym should be introduced later in the abstract or in the text. If this does not work, the title should be modified so that no ambiguity about chemical formulas is present.
Abstracts: line 12 «... simultaneous introduction of 1D and 2D additives were successfully fabricated...» Examples about these kinds of additives are not introduced in the abstract; the reader does not understand from the beginning which is their features, pertinence, and contribution. Instead of using a significant name mainly for the authors, a precise description of these kinds of materials should be provided, while later explanations will allow the introduction of more concise name groups like "1D and 2D additives".
They also are not additives (for example, like additives in polymers), but constituent elements, or even constituent phases.
lines 14-15: « Nanocomposites with SiCw and hBN additives can demonstrate superior hardness, fracture toughness and flexural strength.» This sentence should be the opening of the abstract, because it explains why these materials are interesting; in addition, the mention of SiC/w and h/BN presents them implicitly as 1D or 2D additives.
lines 40-41: «Although the sintering pressure and temperature can be effectively reduced, the metal additives adversely affect the mechanical properties of the composites» the meaning of this sentence is not clear, it should be rephrased.
Materials and Methods, general notes: the following points should be better explained:
how much grams of c/BN were mixed with SiC/w, to produce one or a set of samples? How many samples, in this case?
The amount of given SiC/w is a range, but how many amounts were investigated in this range of SiC/w amount? Not all the data are reported, and references to other conditions are not evident.
Which was the size of the samples? (the size of samples for flexural strength is not good for the other kinds of tests...)
How many samples were produced for each condition?
Which was the sintering atmosphere?
Please, describe better the polishing procedure for the samples
More details are in general needed in the Materials and Method part related to the characterization, that is:
XRD (septime, stepsize, acquisition time, voltage and current applied to the tube, etc.), SEM (filament, applied bias, sample preparation method, as they are possibly not conductive, etc.), EDS (acquisition time, detector, etc.)
Formula 1, please, specify all the measurement units
Figure 1, (a) to (d): «. It is worth noting that no peaks of hBN were detected...» ok, but the indexing is not satisfactory: the reflections are attributed to a phase, but not a crystallographic plane. In addition, no PDF or OCDD files are provided as a reference for the indexing process. Even in this case, the figure needs to be enlarged, because the present phases are barely visible.
Figure 1(a): which is the difference between the pattern of C/BN powders here and of the pattern 0% (0 wt. % of SiC/w, I guess) in figure 2?
How was the average particle size calculated?
lines 119 - 120: «With the increase of SiCw content, the intensity of diffraction peak for SiC increases.» This is not a precise statement, but a very qualitative one. Unless it is quantified , this sentence is trivial and can be omitted, in my opinion.
lines 136-137: «The flexural strength of the cBN-hBN nanocomposites is 486.4 MPa.» Where was this value taken? form the literature? It does not seem that it is coming form figure 3...
lines 141-142: «...highest fracture toughness of 6.2±0.6 MPa.m^(1/2), and flexural strength of 687.4±44.9 MPa.» Where were this values taken? The flexural strength for samples with 25 wt. % SiC/w ~600 MPa, while 6.2±0.6 MPa.m^(1/2) seems to be reasonable for fracture toughness...
Figure 2, EDS: everything needs to be enlarged, as already specified for all the pictures... C distribution seems to follow the same distribution of B and N, and not that one of Si. Is it sure that SiC/w does not decompose, and another CBN rich phase is not formed?
Figure 3: why is there a abrupt diminution of strength for SiC/w > 25 wt. %?
line 167 and following: «axial direction could also be clearly observed,» please, put in evidence the axial direction in figure 4c.
«nanoscale crack-deflection and grain-bridging phenomena» please put them in evidence in figure 4c.
«arranged axially/radially» what do the author mean with axially and radially?
Table 1: as already pointed out previously, it is not clear where the value 687.4 MPa for flexural strength was found, because it does not correspond with the data reported in Figure 3
Another table with crystallographic features of the phases, and what was found in TEM analyses should be presented. This should also comprise reference files (PDF, ICDD, etc.), as above highlighted.
Figure 4 caption: «presence of the crack deflection and crack bridging.» Please, show the two phenomena in the picture.
Conclusions: «simultaneous introduction of 1D and 2D additives were fabricated by high-pressure sintering mixed cBN and SiCw» it should be stated clearly in the text that SiCw is a "1D" component, while h/BN is the "2D" component. They are not additive, but components or phases. h/BN is not introduced, but it is formed from c/BN with the addition of SiC/w (why? please explain)
«Composites with 1D SiCw and 2D hBN...» ok, but it should be made clear from the beginning
Reviewer 4 Report
The manuscript entitled "Synthesis of cBN-hBN-SiCw nanocomposite with superior hardness, strength, and toughness" presents an interesting study on the synthesis in one step of a nanocomposite containing cBN and hBN with SiC as a whisker. The present work is based on a good experimental design.
However, there are some issues that should be addressed.
The Abstract Section - as here are first introduced the nanocomposites, please do not use the acronyms directly.
The Introduction Section should be extended as there is extensive literature on nanocomposites containing cBN and SiC.
Line 14 - "The hBN was in-situ introduced..." correctly is to consider as in-situ generated, not introduced!
Line 14-15 - "Nanocomposites with SiCw and hBN additives can demonstrate..." the usage of the modal verb "can" in this context is not appropriate, better reformulate.
Line 19-20 - The allegations are not sustained by the extended content of the manuscript. Please explain these aspects in a thorough manner within the Discussion Section.
Line 77-78 - "SiCw with different contents (5 wt.%– 30 wt.%) were mixed..." The understanding of the phrase is that SiCw has different contents of something! It is necessary to reformulate the phrase.
Line 104 - For equation (1) there were not explained all the parameters, i.e. H, n, or the significance of the constant 0.16. Also, the measuring units should be specified where it applies.
Line 110 - How was it calculated the particle dimensions of 592 nm? From diffractograms or from SEM micrographs?
Line 119 - "...no reaction between cBN and SiCw...." To what reaction are you referring?
Line 118-119 "... there is the partial transformation from cBN to hBN..." The XRD results show that only a very reduced amount of cBN was transformed in hBN! There was not a partial transformation that would support the subsequent allegations formulated in the following paragraphs! Please explain!
Line 185-187 - "The introduction of hBN by in situ phase transformation ensures that there is a uniform distribution of 2D materials between ceramic grains, which is a remarkable advantage over other mixing methods." Again, please revise this formulation of "introduction in situ"! Is to be discussed to which extent the hBN could be uniformly distributed between ceramic particles as time as the XRD showed such a limited transformation cBN into hBN phase!
Line 191- "it can be observed that cBN and SiC form a good bonding interface." Please be consistent in your affirmations. In the previous Line 119 it was mentioned that "no reaction" occurred between cBn and SiC, and in Line 191 it is mentioned a good bonding - as a consequence, some more explanations are needed here.
Line 201 - Figure 5 - the images presented are very interesting. However, it seems that there is presented only one section where it was found hBN, and it is not found the same structural aspect over the whole surface at the interface. Please include the appropriate explanation in such a way as to balance the XRD results where only a faint quantity of hBN is obtained and the images from Figure 5 that shows a significant presence of hBN.
Line 214-215 - "Our work devoted to preparing nanocomposites by introducing multiple additives through various means..." In the presented manuscript, only SiC was introduced as a whisker, while hBN was generated in situ, so to which multiple additives are you referring?
Round 2
Reviewer 3 Report
REVIEW ROUND #2:
Paper «Synthesis of cBN-hBN-SiCw nanocomposite with superior hardness, strength, and toughness» - manuscript ID: nanomaterials-2049461
The paper improved, the use of English is almost fine, but an unresolved issue still remains:
Example of English expressions to be fixed (not exhaustive list):
line 56 and following: «One-dimensional (1D) materials involving nanofibers [20, 26, 27], nanotubes [21, 22, 28] and whiskers [23-25, 29], which are singular structures with the lateral dimension in submicron or nanometer scale.» a finite tense is missing
line 97: «with Φ 5 ×5 mm» it should be "with Φ (diameter) = 5 mm × 5 mm"
The relevant point:
please, index plains in XRD diffraction patterns; the size is fine, but reflections should be indexed. Thank you
Reviewer 4 Report
The authors significantly improved their manuscript.
Although, the Introduction section could be improved better.
